# Work–Family Interaction, Self-Perceived Mental Health and Burnout in Specialized Physicians of Huelva (Spain): A Study Conducted during the SARS-CoV-2 Pandemic

**DOI:** 10.3390/ijerph19063717

**Published:** 2022-03-21

**Authors:** Francisco-Javier Gago-Valiente, Emilia Moreno-Sánchez, Alba Santiago-Sánchez, David Gómez-Asencio, María-de-los-Ángeles Merino-Godoy, Estefanía Castillo-Viera, Emília Isabel Costa, Adrián Segura-Camacho, Luis-Carlos Saenz-de-la-Torre, María-Isabel Mendoza-Sierra

**Affiliations:** 1Area of Preventive Medicine and Public Health, University of Huelva, 21007 Huelva, Spain; francisco.gago@dstso.uhu.es; 2Department of Pedagogy, Faculty of Education, Psychology and Sports Sciences, University of Huelva, 21071 Huelva, Spain; emilia@dedu.uhu.es (E.M.-S.); david.gomez419@alu.uhu.es (D.G.-A.); 3Health Center Cazalla de la Sierra, 41370 Seville, Spain; alba.santiago764@alu.uhu.es; 4Nursing Department, Faculty of Nursing, University of Huelva, 21007 Huelva, Spain; 5Integrated Didactics Department, Faculty of Education, Psychology and Sports Sciences, University of Huelva, 21071 Huelva, Spain; estefania.castillo@dempc.uhu.es; 6Health Sciences Research Unit: Nursing, 3000 Coimbra, Portugal; eicosta@ualg.pt; 7Nursing Department, Health School, University of Algarve, 8000 Faro, Portugal; 8Department of Social, Development and Educational Psychology, Faculty of Education, Psychology and Sports Sciences, University of Huelva, 21007 Huelva, Spain; adrian.segura@dpee.uhu.es (A.S.-C.); luis.saenz@dpces.uhu.es (L.-C.S.-d.-l.-T.); imendoza@dpsi.uhu.es (M.-I.M.-S.)

**Keywords:** work–life balance, burnout, coronavirus infections, mental health, medical staff, physician, nursing, public health

## Abstract

Background: The medical staff who work in specialized healthcare are among the professionals with a greater risk of presenting negative indicators of mental health. These professionals are exposed to numerous sources of stress that can have a negative influence on their personal life. Currently, SARS-CoV-2 poses an additional and relevant source of stress. The aim of this study was to identify the interactions between the work and family environments, as well as to analyze self-perceived mental health and burnout in physicians who, during the COVID-19 pandemic, carried out their jobs in public health in Huelva (Spain), also considering a series of sociodemographic variables. Methods: This is a descriptive, cross-sectional study. Information from 128 participants was collected using the SWING, MBI-HSS and GHQ-12 questionnaires, along with sociodemographic data and possible situations of contact with SARS-CoV-2. The data were analyzed, and correlations were established. Results: Most of the sample obtained a positive interaction result of work over family. Those who had been in contact with SARS-CoV-2 represented higher percentages of a positive result in GHQ-12, negative work–family interaction, burnout, emotional exhaustion and depersonalization. In general, the men showed a worse mental health state than women. Conclusions: The medical staff of Huelva who had been in contact with situations of SARS-CoV-2 in their work environment presented worse indicators of mental health and greater negative interaction of work over family than those who had not been in contact with these situations.

## 1. Introduction

In the last three decades, the number of people who currently reconcile work and family responsibilities have increased significantly in Spain. This increase has resulted in a larger number of single-parent families, a higher percentage of women in the world of work and business, the existence of couples with different working schedules and parents who are increasingly involved in the upbringing of their children, as well as an increase in housework co-responsibility [1]. Some important sociological changes have caused a crisis in the traditional family model, which has forced a large proportion of the population to make great efforts in order to balance the two most important spaces of life: family and work [2].

Currently, due to the incompatibility and low flexibility of working schedules, leisure and family time has been reduced; therefore, almost one third of the Spanish population claim to have problems coordinating their work and family lives [3].

The professional discomfort derived from this context can have a negative influence on the family environment, thereby raising the stress levels. Moreno, Sanz, Rodríguez and Geurts [4] defined the work–family interaction as a series of work and family demands that are reciprocally incompatible, thus generating conflicts in the person; they classified it in 4 dimensions: “positive work–family interaction, negative work–family interaction, positive family–work interaction and negative family–work interaction” (p. 34). Therefore, reconciliation between work and family lives is not always effective. The reciprocal influence between the work and non-work contexts can result in either a situation of balance or a conflict/interference; the latter occurs if the influence is negative, with the possibility of generating harmful consequences, such as anxiety or burnout, in people who go through this type of experiences [5].

Chronic occupational stress has been recently recognized and categorized by the World Health Organization (WHO) as burnout syndrome [6]. The experience of burnout has been related to a long list of negative personal, social and organizational results. This syndrome consists of three dimensions: high emotional exhaustion, high depersonalization, and low personal realization, which are indicators of mental health [7].

In certain occupations with special characteristics, such as medicine, there are some barriers to the effective application of reconciliation measures [8]. This poses an additional risk factor of occupational stress in this group, who, due to the characteristics of their clinical-healthcare job, are already among those with greater probabilities of presenting negative indicators of mental health [9]. Moreover, burnout usually appears in healthcare staff with effects such as poor quality in the care of patients and an increase of medical errors [10].

In addition to these concerning aspects, the effects of the health alert generated by the COVID-19 pandemic worsen the psychological discomfort of many workers, especially in healthcare professionals [11,12].

The correlation between professional discomfort in healthcare professionals and pandemic states has already been demonstrated in previous studies, which showed that such workers feared contagion and infecting their relatives, friends, and co-workers [13], and they also felt uncertainty and stigmatization [14,15] which could result in psychological consequences in the long term [16].

During the last two years, several reports have been published about the effects of the SARS-CoV-2 pandemic on the mental health of healthcare workers [17,18]. However, all the information is important, as it allows a more comprehensive analysis of this problem: the impact of the pandemic on the well-being of professionals who are on the front line of healthcare. Therefore, a study was designed to identify the interactions between the work and family environments, as well as to analyze the self-perceived mental health and prevalence of burnout, in specialized physicians who carried out their job during the pandemic state in public health care in Huelva (Spain). The relationship between these dependent variables was also analyzed, considering a series of sociodemographic variables, such as age, sex, years of professional experience and parent situation.

The results of this study are presented according to the model of Golembiewski and Munzenrider [19], where the classification of the worker is rather based on the virulence of the syndrome than on the unit or service to which the worker belongs. That is, the results are presented according to the affectation of the syndrome.

## 2. Materials and Methods

### 2.1. Participants

This is a descriptive, cross-sectional study with a sample of 128 medical professionals of a hospital in Huelva city (Spain). The distribution with respect to marital status, sex and parenthood was very heterogeneous (Table 1).

### 2.2. Procedure

The information was collected in the participating hospital of Huelva (Spain). The field work was carried out in April, May and June 2020. Figure 1 shows the evolution of the number of cases of SARS-CoV-2 confirmed by active infection diagnostic test (AIDT) in the different districts of the province of Huelva.

The probabilistic model was used to select the sample, with a sampling error of 0.05. A work schedule was designed and different visits to the hospital were performed in the different morning and evening shifts, in different services and units, in order to avoid duplicated information. In each visit, a 30-min questionnaire was administered to each participant in paper format and in the presence of the researcher or the collaborator to make the relevant clarifications. All participants signed the written informed consent, which guaranteed the anonymity of the data. The response rate was 75.1%. The main reasons stated to justify a refusal to complete the questionnaire were work load and lack of time. The director of the hospital authorized the collection of data, and the study was approved by the Research Ethics Committee.

### 2.3. Instruments

The participants complete a brief sociodemographic questionnaire, providing information about their age, sex, service time, marital status and parent condition. An item was introduced at the end of this questionnaire, which asked the participants whether they had been in contact with someone in a situation of SARS-CoV-2 in the workplace (Appendix A).

One of the instruments used was the Work–Home Interaction Survey—Nijmegen (SWING) (Appendix A), which was developed by Geurts, Taris, Kompier, Dikkers, Van-Hooff and Kinnunen [21] to measure the negative and positive relationships between work and family. It consists of 22 Likert items with 4 points (0 = never, 1 = sometimes, 2 = often, 3 = always). This questionnaire is used to identify the type of work–family and family–work relationship of workers. We used the Spanish-validated version [4]. These authors analyzed the psychometric properties of the Spanish version of SWING in a sample of 283 emergency workers. The data of the confirmatory factor analysis indicated that the four-factor model with no correlation between the components of positive and negative interaction was the one that best fitted the obtained information. The original format of the instrument was maintained, in which the items are distributed in four sub-scales: negative work–family interaction, negative family–work interaction, positive work–family interaction and positive family–work interaction. The analysis of the reliability of the scale revealed that this Spanish version has an adequate internal consistency, with values between 0.77 and 0.89. Significant correlations were found between SWING and variables related to work, family, and well-being, which contributes to its convergent validity. It is thus concluded that this version presents adequate psychometric properties for the present study.

The classification of each participant into one or another subcategory was based on the score obtained in each of the latter, falling into the one where they obtained the highest score. For negative work–family interaction, items 1–8 were added and divided by 8; for negative family–work interaction, items 9–12 were added and divided by 4; for positive work–family interaction, items 13–17 were added and divided by 5; and, for positive family–work interaction, items 18–22 were added and divided by 5.

The possible non-psychotic psychiatric pathologies (self-perceived mental health) were evaluated through the short version (12 items) of the General Health Questionnaire (GHQ-12) (Appendix A). This instrument was designed to identify possible non-psychotic psychiatric pathologies in the general population [22]. We considered the findings of the Spanish validation studies and the suggestions of the authors of the questionnaire to select a cutoff point of ≥12 in the classification of people who could have mental or emotional disorders [23,24]. GHQ-12 has been validated in Spain and applied in numerous studies to evaluate the general population [22,24,25]. Another investigation [26] conducted studies to validate the questionnaire in a sample of 1641 people, obtaining an adequate internal consistency, with general Cronbach’s alpha results of 0.90. In addition, studies carried out in other countries [27] also reported adequate psychometric and reliability properties for the instrument in a population of 854 participants, with a Cronbach’s alpha of 0.80.

Lastly, the Maslach Burnout Inventory (MBI), in its general version developed by Maslach and Jackson [7], which has been validated in the international context, was also used. We applied the version designed for healthcare professionals, that is, the MBI—Human Services Survey (MBI–HSS) (Appendix A). The use of this questionnaire for the identification of burnout was motivated by its reliability indices in the different dimensions of the syndrome (0.90 for emotional exhaustion, 0.71 for personal realization, and 0.79 for depersonalization, with an internal consistency of 0.80 for all items). Similarly, to validate the questionnaire, factor analyses were carried out, which required a three-dimensional structure that corresponded to the dimensions of the burnout syndrome; that is, the scale measures exactly what the study variable was desired to measure [28]. This type of factor validity is supported by convergent validity studies performed by the authors of the present study. There are also studies that analyzed MBI, obtaining a Cronbach’s alpha of 0.78 for emotional exhaustion, 0.71 for depersonalization and 0.76 for personal realization [29]. Therefore, the efficacy of MBI for the present study has been objectified.

The level of affectation was evaluated in the different dimensions using a 7-point (0 to 6) classification scale. The higher the score, the greater the level of affectation for emotional exhaustion and depersonalization. However, for personal realization, the lower the score of its items, the greater the affectation of the person in this dimension. Thus, the cutoff point to categorize people with high emotional exhaustion was ≥27 points in the sum of the items of this dimension within the questionnaire, ≥10 points for high depersonalization, and ≤33 points for low personal realization [7]. The diagnosis of a person with burnout syndrome is established when there is an affectation in all three dimensions, that is, high emotional exhaustion, high depersonalization, and low personal realization. Thus, based on the three subscales or dimensions, the results are categorized in high, medium, and low levels, and the scores are indicated below:Emotional exhaustion: high (≥27), medium (19–26) and low (≤18).Depersonalization: high (≥10), medium (6–9) and low (≤5).Personal realization: high (≥40), medium (34–39) and low (≤33).

### 2.4. Data Analysis

The statistical analysis was carried out using SPSS software (Statistical Products and Service Solutions) v.23.0.

Firstly, a univariate analysis was performed, calculating the standard deviation, mean, and minimum and maximum values of the quantitative variables.

Secondly, we calculated the percentages and frequencies of the following variables: marital status, sex, work–family interaction, parenthood, burnout, and its dimensions (emotional exhaustion, depersonalization and personal realization), self-perceived general health (probable or non-probable non-psychotic psychiatric case) and contact with situations of SARS-CoV-2.

Moreover, we included normality tests for the quantitative variables to obtain information from subsequent hypothesis tests, that is, whether parametric or non-parametric tests should be used. Since the amount of data used was above 50, we decided to use the Kolmogorov–Smirnov statistic for the normality tests.

Then, the following statistical analyses were carried out based on the aim of the study:Cross-tabulation analysis for work–family interactions as a function of sex, burnout, or affectation of its dimensions (emotional exhaustion, depersonalization and low personal realization) and contact with SARS-CoV-2. Moreover, Chi-squared tests were conducted between these variables.Cross tabulations for burnout, emotional exhaustion, depersonalization, and personal realization as a function of sex, marital status, situations of contact with SARS-CoV-2 and parenthood. Moreover, Chi-squared tests were conducted between these variables.Since the normality tests for service time, emotional exhaustion (quantitative), depersonalization (quantitative) and personal realization (quantitative) showed an abnormal distribution, we used the Mann–Whitney U-test for the independence tests with categorical variables of two categories and the Kruskal–Wallis H-test for 3 or more groups. Correlations were also analyzed between the different study variables through Spearman’s Rho.Cross tabulations for possible or non-possible non-psychotic psychiatric cases in relation to burnout, emotional exhaustion, depersonalization, personal realization and contact with SARS-CoV-2. Moreover, Chi-squared tests were conducted between these variables.Cross tabulations as a function of the results obtained in GHQ-12 (possible or non-possible non-psychotic psychiatric case) in relation to sex, marital status, and parenthood. Chi-squared tests were conducted between these variables. Moreover, the Mann–Whitney U-test was used in the hypothesis test for the variables of results obtained in GHQ-12 and service time.

## 3. Results

### 3.1. Work–Family Interaction

It was shown that 48.9% of the sample presented a negative work–family interaction, 6.2% presented a positive work–family interaction and 44.9% presented a positive family–work interaction. No participant presented a negative family–work interaction.

### 3.2. Probable Non-Psychotic Psychiatric Pathologies (GHQ-12)

Figure 2 shows that most of the participants did not present possible non-psychotic psychiatric pathologies. The percentage of affectation of women was slightly higher than that of men.

### 3.3. Prevalence of Burnout, High Emotional Exhaustion, High Depersonalization and Low Personal Realization

Table 2 shows the prevalence of the syndrome and that of its dimensions. As can be observed, there is a very low percentage of professionals with burnout. The most affected dimension was depersonalization and the least affected personal realization. Although there is a low percentage of burnout affectation, the percentages of affectation of the variables are important. In the following sections of the manuscript, the dependence with other variables will be analyzed.

### 3.4. Correlation of Work–Family Interaction with Sex, Situations of Contact with SARS-CoV-2, Burnout, Emotional Exhaustion, Depersonalization, and Personal Realization

Women showed a higher percentage of positive work–family interaction (11.2%) than men (3.3%). On the other hand, the percentage of men was higher in the negative work–family interaction (43.7%) compared to women (32.4%). In the positive family–work interaction, both sexes obtained similar percentages, with 56.4% of women and 53.1% of men. There was dependence between both variables (X^2^ = 19.757; *p* = 0.000 at 95% CI).

The analysis of the interactions of work over family and vice versa as a function of contact or no contact with SARS-CoV-2 revealed a higher percentage of negative work–family interaction in people who had been in at least one of such situations (47.1%) compared to those who never had contact with COVID-19 patients (19.5%). In the positive work–family interaction and positive family–work interaction, there were higher percentages for the participants who had not been in contact with SARS-CoV-2 (9.4% and 71%, respectively) than for those who had (9.2% and 43.7%, respectively). In the study of the possible relationship between the two variables, dependence was revealed (X^2^ = 86.190; *p* = 0.000 at 95% CI).

Lastly, the analysis of the relationship between the different interactions of work over family with burnout syndrome and its dimensions showed dependence between all these variables at a 95% confidence interval. Thus, the participants who suffered from burnout and high affectation of emotional exhaustion and depersonalization also obtained higher percentages of negative interactions of work over family. However, the participants with greater affectation in their personal realization (low personal realization) showed a higher percentage of positive interactions of family over work (Table 3).

### 3.5. Correlation of Burnout and Its Dimensions with Sex, Marital Status, Contact with Situations of SARS-CoV-2 and Parenthood

In the analysis of the relationship between burnout and sex, the *p*-value of the Chi-squared independence test was significant (X^2^ = 31.795; *p* = 0.000 at 95% CI). Therefore, there is a dependence between these two variables. Men presented higher percentages of burnout (17.50%) than women (6%) (Figure 3).

Dependence was also observed between the dimensions of the syndrome and sex ([X^2^ = 10.320; *p* = 0.006 for emotional exhaustion]; [X^2^ = 58.728; *p* = 0.000 for depersonalization] and [X^2^ = 19.419; *p* = 0.000 for personal realization] at 95% CI). Men presented higher percentages of affectation in high emotional exhaustion and high depersonalization (44.5% and 40.8%, respectively) than women (41.8% and 26.9%, respectively). In low personal realization, men also obtained higher percentages of affectation (34.1%) than women (22%) (Figure 3).

Regarding the relationship between burnout and marital status, dependence was observed between these two variables (X^2^ = 27.245; *p* = 0.000 at 95% CI). The divorced participants presented the highest percentages of burnout (23.5%), whereas the lowest percentages corresponded to the widowed participants (0%).

Dependence was also observed between contact with situations of SARS-CoV-2 and burnout (X^2^ = 44.514; *p*= 0.000 at 95% CI). The participants who had been in contact with situations of SARS-CoV-2 in their workplace presented higher percentages of burnout (89.2%) than those who had not (10.8%) (Figure 4).

No dependence was shown between burnout and parenthood (X^2^ = 0.001; *p*= 0.978 at 95% CI).

The dimensions of burnout presented dependence with marital status and contact with situations of SARS-CoV-2 (at 95% CI). However, no dependence was observed between the dimensions of the syndrome and the condition of having children (at 95% CI). Table 4 shows the different percentages of affectation and the value of the indices of the analyses performed.

### 3.6. Correlation of the Results in GHQ-12 (Possible or Non-Possible Non-Psychotic Psychiatric Case) with Burnout, Emotional Exhaustion, Depersonalization, Personal Realization and Contact with Situations of SARS-CoV-2

Dependence was observed between the result in GHQ-12 and burnout (X^2^ = 70.564; *p* = 0.000 at 95% CI). Thus, the participants who had possible non-psychotic psychiatric pathologies also represented higher percentages of burnout (89.2%) than those who obtained a negative result in GHQ-12 (10.8%) (Figure 5).

Similarly, there was dependence between the result in GHQ-12 and high emotional exhaustion and high depersonalization ([X^2^ = 167.362; *p* = 0.000] and [X^2^ = 30.645; *p* = 0.000] at 95% CI). The participants who obtained a positive result in GHQ-12 also presented higher percentages of high emotional exhaustion and high depersonalization (58.7% and 37.2%, respectively) than those who obtained a negative result (28.5% and 25%, respectively) (Figure 5).

Personal realization also showed dependence with the result in GHQ-12 (X^2^ = 21.725; *p* = 0.000 at 95% CI). In contrast with the other two variables, the participants with a positive result in GHQ-12 presented higher percentages of high personal realization (31.5%) than those with a negative result (19.5%) (Figure 5).

Lastly, dependence was observed between the result in GHQ-12 and contact with situations of SARS-CoV-2 (X^2^ = 62.483; *p* = 0.000 at 95% CI). The participants with non-psychotic psychiatric pathologies also presented higher percentages of contact with situations of SARS-CoV-2 in their workplace (58.7%) than those with a negative result in GHQ-12 (41.3%) (Figure 5).

### 3.7. Correlation of the Result in GHQ-12 with Sex, Marital Status and Parenthood

There was no dependence between the result in GHQ-12 and sex and parenthood ([X^2^ = 0.076; *p* = 0.782] and [X^2^ = 1.909; *p* = 0.167], respectively, at 95% CI).

Dependence was obtained between the result in GHQ-12 and marital status (X^2^ = 23.588; *p* = 0.000 at 95% CI). Figure 6 represents the percentages of responses as a function of marital status. The widowed participants presented the highest percentages of the probable non-psychotic psychiatric case (64.30%), whereas the lowest percentages corresponded to the participants with a partner (32%).

### 3.8. Correlation of the Result in GHQ-12, Burnout, Emotional Exhaustion, Depersonalization and Personal Realization with Service Time

Mann–Whitney U-tests were carried out for the result in GHQ-12 and service time. No statistically significant differences were observed between the participants with possible non-psychotic psychiatric pathologies and those with a negative result in GHQ-12 as a function of service time (Mann–Whitney U-test = 123951; *p* = 0.775 at 95% CI).

Statistically significant differences were observed in burnout with respect to service time (Mann–Whitney U-test = 32591; *p*= 0.000 at 95% CI). The average number of service years of participants with burnout was lower (14.5 years) than that of those without burnout (17.8 years).

No statistically significant differences were detected in the affectation of personal realization as a function of service time (Kruskal–Wallis H = 2.094; *p* = 0.622 at 95% CI).

Statistically significant differences were obtained in emotional exhaustion and depersonalization as a function of service time (Kruskal–Wallis H = 28.557 and 22.959, respectively, *p* = 0.000 for both at 95% CI). However, the correlation between the two dimensions with service time was positive, although very weak (Table 5).

## 4. Discussion

The results present a higher incidence in the positive interaction of family over work in both men and women. These data are in line with those of previous studies [30] that analyzed the relationship between work commitment and family–work interaction in healthcare professionals. Such studies reported a positive interaction of family over work, i.e., an enrichment of family over work in professionals who presented a high work commitment. Moreover, other studies highlighted a higher percentage of men than women with a negative interaction of work over family, and a higher percentage of women with a positive interaction [31]. It is important to highlight that these differences are not observed in all studies since there are investigations in which no statistically significant differences have been found in the interactions as a function of sex [32]. On the other hand, some studies show that men could have more difficulties accessing measures of organizational reconciliation than women [33].

The percentages obtained in the SWING scale (work–family interaction) indicate that most of the population had a positive interaction of family over work, where, in addition, the participants who did not have children presented, in general, worse self-perceived health than those who did have children. Therefore, the family environment could be acting as a modulating variable in the mental health of many professionals who did not show probable non-psychotic psychiatric pathologies, in the period of the present study, during the pandemic [34,35].

Regarding the type of interaction between work and family as a function of contact with SARS-CoV-2, the data are in line with those of previous studies about the impact of the SARS-CoV-2 pandemic on healthcare professionals, which report discomfort in this population, where one of the factors is the fear of infecting their relatives [17]

The conclusions of other works [36,37,38] agree with many of the findings of the present study. It is worth highlighting the dependence between sex and affectation of emotional exhaustion and depersonalization, with women presenting lower levels of affectation in both dimensions. Other studies [39] attribute and relate high emotional exhaustion to low personal realization, with men being more affected than women. These sex differences could be due to the connection between gender (and the culturally assigned roles) and other characteristics and sociodemographic variables. Belonging to a certain sex could involve a higher frequency in the presence of other variables that would act as modulators or enhancers of emotional exhaustion and depersonalization [34].

Furthermore, the information provided in numerous studies that relate the service time to emotional exhaustion or depersonalization show contradicting results, since the relationship that was established between service time and these variables was direct for some authors [40], and not so direct for others [41]. Despite the information obtained in these studies, it is important to take into account that the results regarding service time must be considered always with caution in this topic, due to the probable survival bias; that is, those who suffer from early burnout in their careers are likely to quit their jobs, leaving behind those who stay, who, consequently, show lower burnout levels [24].

In the study of the relationship of sociodemographic variables, such as marital status, some investigations [42] present higher levels of emotional exhaustion or depersonalization in married people or people with partners, whereas others report higher levels in single people [43]. According to the results obtained in the present study, it is considered that analyzing marital status as the only influence of family over work could have a series of biases. Family support does not necessarily have to come from the partner, since a person can receive support from their parents, nephews/nieces, children, cousins, or other relatives.

With respect to the participants who had children, although they could be expected to present a negative influence in their professional career and, therefore, in their personal realization due to the lack of time, as has been stated in previous studies [44], the population of the present study did not provide any results in this regard. It would be interesting to determine whether physicians feel satisfied with the family reconciliation measures received.

There are also professionals who, despite presenting high emotional exhaustion, high depersonalization and poor general self-perceived health, show high personal realization, since the attainment of goals allows them to overcome challenges at work [45]. This could cause these symptoms to go undetected in people who suffer from them.

The obtained results in terms of GHQ-12 score, in relation to marital status and parenthood, also show that the family environment could be acting as a modulating variable in the mental health of these professionals, as these symptoms went undetected in the participants who suffered from them [34,35]. Some studies describe more mental health problems in people who do not have a partner compared to those who do have a partner [43].

Furthermore, the relationship detected between depersonalization and the positive result in GHQ-12 provides relevant data that must be considered. Depersonalization can be a symptom of major depression. Previous studies have reported that in patients with unipolar depression, the symptoms of associated depersonalization are more intense compared to the healthy control population and that there is also a positive correlation between depression and depersonalization [46]. Perhaps these affectations are not discrete categories, but they may have common biological origins and may be at least part of one continuous spectrum or affective disorders [18].

Various studies have been published in which an analysis is made of the effect of the pandemic on the mental well-being of health professionals, however, as it is still an active situation and whose impact is expected to last over time, we consider this analysis pertinent. The literature on this topic provides information that agrees with the results of the present study. The data reveal high percentages of healthcare professionals who had been in contact with a situation of SARS-CoV-2 and suffered from symptoms of anxiety, stress, depression and sleep disorders [12,17,47].

Emotional exhaustion and depersonalization already presented greater affectation in men than in women before the COVID-19 pandemic [48]. However, the percentages of the population with an affectation of both variables were higher in the present study conducted during the pandemic [47,49].

Lastly, with respect to self-perceived mental health as a function of sex, the results are in line with those of other studies [50], which reported that women perceived a worse mental health state than men. In contrast with these studies, the present work did not detect dependence between sex and the result in GHQ-12.

### 4.1. Applications to the Clinical Practice

The results of this investigation show the convenience of developing mitigation strategies in pandemic situations. Since resources can be particularly scarce during these states, adequate psychological support could be provided in many different forms, including telemedicine and informal support groups [34].

Medical staff play a fundamental role in the mental healthcare of patients. In most cases, they provide the first psychological aid, since they are the first professionals who attend to the demands of their patients after the clinical diagnosis, and they provide emotional support in states of convalescence to the worried relatives of the patient [51]. Therefore, it is fundamental for these professionals to maintain a good mental health state in order to provide quality healthcare.

It is important to highlight that the data of the present study corroborate the need of considering gender as a key for a change of paradigm that will allow generating more accurate and inclusive scientific knowledge, as well as a more realistic, fair and egalitarian healthcare system.

### 4.2. Limitations

Despite the richness and impact of the obtained data, this study presents certain limitations that must be overcome, and which constitute strengths of the origins of this study. Since this is a cross-sectional study, which measures the relationship between the variables at a specific period without conducting a follow-up in time, many of these variables may be altered, as in the case of the Spanish public healthcare system [52], and such changes can modify the scores of many objective and subjective indicators of health.

Moreover, it would also be convenient to delve into the possible psychiatric history of the participants and include items in the questionnaires about access to resources (social, economic, healthcare, etc.) during the pandemic. This pandemic has had a greater impact on lower socioeconomic levels. Therefore, it would be very interesting to perform prospective studies, with the same sample, including new items with more variables, in order to reevaluate the situation and the possible changes.

Another limitation is related to the sample size. Although the number of healthcare professionals did represent the reality of this occupational group in the participating hospital, the results cannot be extrapolated to other hospitals or provinces.

Thus, it is necessary to carry out new studies with larger sample sizes to obtain results with greater external validity, including more healthcare centers and classifying their medical staff according to their different services and units. It would also be convenient to conduct correlation studies between the result in GHQ-12 and emotional exhaustion, depersonalization and personal realization.

## 5. Conclusions

We can conclude that the medical staff of the hospital of Huelva (Spain) who had been in contact with situations of SARS-CoV-2 in their work environment presented a worse mental health state than those who had no contact with such situations. This state was demonstrated through high emotional exhaustion, high depersonalization and a positive result in probable non-psychotic psychiatric pathologies. There is also dependence between these variables.

Secondly, it is worth mentioning that those participants who had been in contact with situations of SARS-CoV-2 in their work environment also presented higher percentages of negative interactions of work over family than those who had had no contact with such situations.

Lastly, there is a difference in the affectation of some variables as a function of sex. Men presented, in general, a worse mental health state than women.

### Relevance for Clinical Practice

Regarding strengths, this study provides interesting information about a series of mental health indicators in medical staff, thereby updating the knowledge on this topic to contribute to future intervention lines that address this problem. A possible roadmap is thus set for future larger studies in the same line.

Another relevant strength of this study was the evidence in the different affectation of the mental health state as a function of sex in the obtained results, which corroborates the importance of considering gender in health sciences research.

Thus, the analysis of the findings of this study underlines the imperative need to enable results to be used with a psychosocial and educational approach to modify vulnerable targets through the design and implementation of training and preventive programmes.

## Figures and Tables

**Figure 1 ijerph-19-03717-f001:**
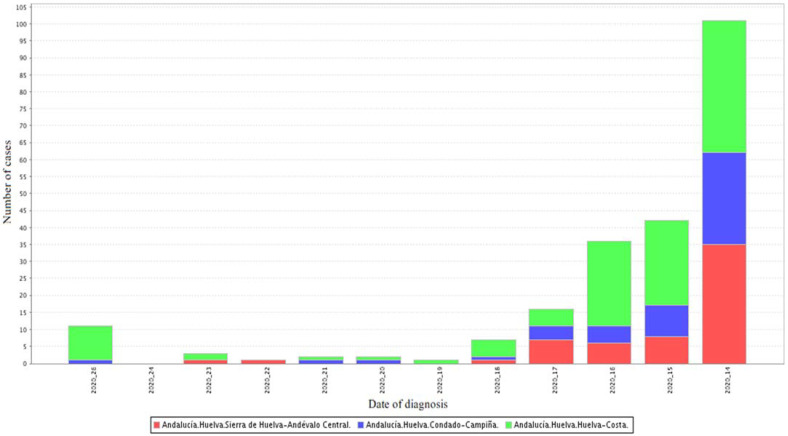
Cases of SARS-CoV-2 in the different districts of the province of Huelva grouped by weeks [20].

**Figure 2 ijerph-19-03717-f002:**
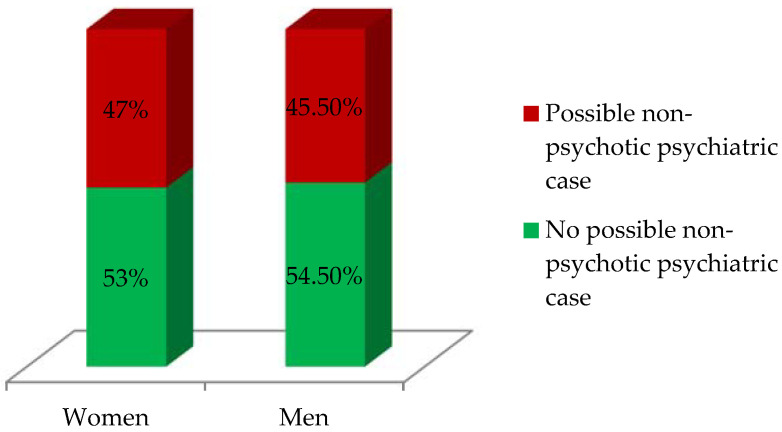
Responses to GHQ-12 grouped by sex.

**Figure 3 ijerph-19-03717-f003:**
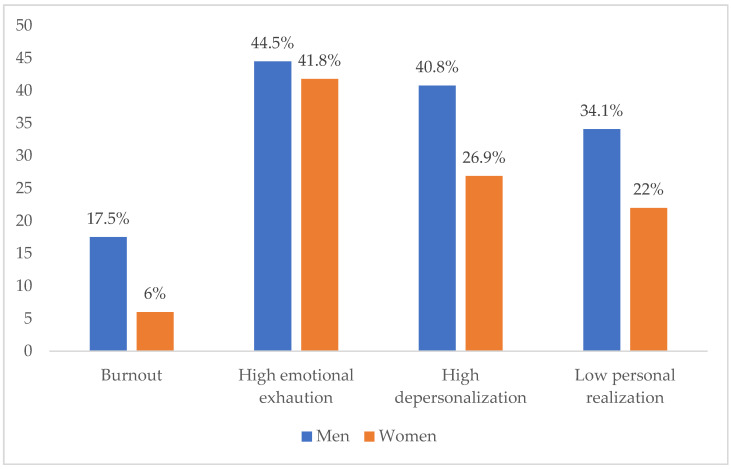
Correlation of burnout and its dimensions with sex.

**Figure 4 ijerph-19-03717-f004:**
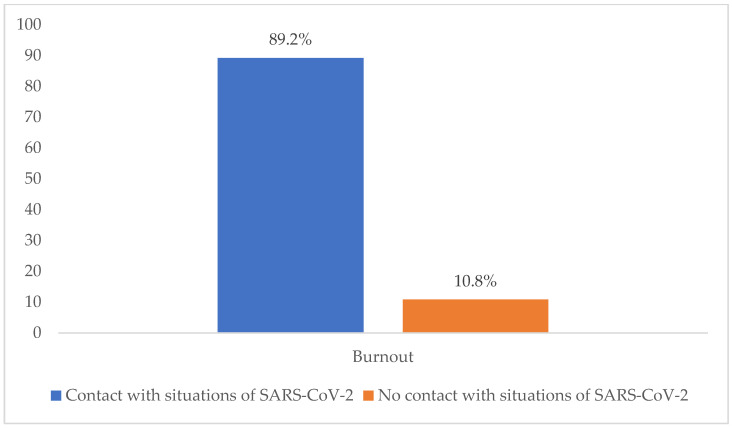
Correlation of burnout and its dimensions with contact with situations of SARS-CoV-2.

**Figure 5 ijerph-19-03717-f005:**
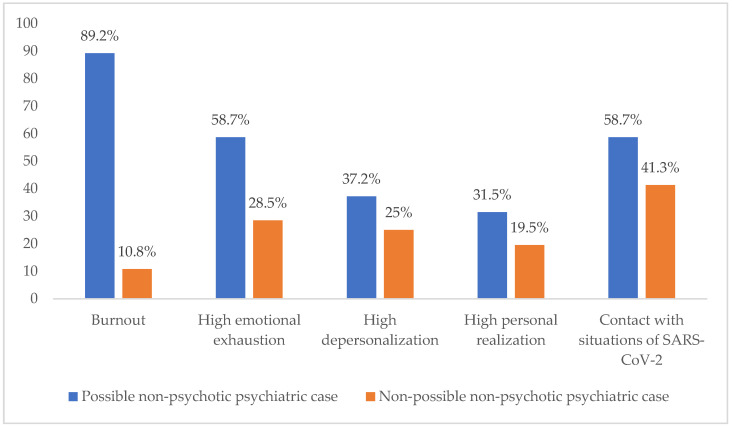
Correlation of the results in GHQ-12 with burnout, emotional exhaustion, depersonalization, personal realization and contact with situations of SARS-CoV-2.

**Figure 6 ijerph-19-03717-f006:**
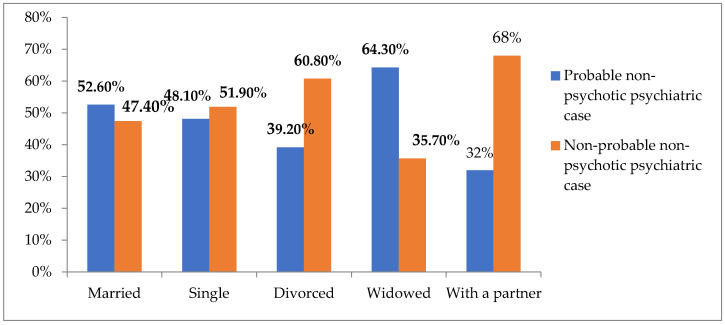
Results in GHQ-12 as a function of marital status.

**Table 1 ijerph-19-03717-t001:** Distribution of sociodemographic variables.

Sociodemographic Variables				
		%	*n*	Age
Sex	Men	17.18%	22	Between 27 and 63 years
	Women	82.81%	106	Between 29 and 64 years
Parenthood		Yes (%)		No (%)
		64.10%		35.90%
Marital status		%		
	Married	46.70%		
	Single	30.90%		
	Divorced	4.80%		
	Widow	3%		
	With a partner	14.60%		

**Table 2 ijerph-19-03717-t002:** Percentages of burnout, high emotional exhaustion, high depersonalization and low personal realization.

Variables	Burnout		Emotional Exhaustion			Depersonalization			Personal Realization		
	Yes	No	High	Medium	Low	High	Medium	Low	High	Medium	Low
Percentages (%)	5.1%	94.9%	28.7%	41%	30.3%	31.5%	18%	50.6%	33.7%	46.1%	20.2%

**Table 3 ijerph-19-03717-t003:** Group statistics and Pearson’s Chi-squared test for burnout and its dimensions with the dimensions of SWING.

Dependent Variable		Negative Work–Family Interaction	Positive Work–Family Interaction	Positive Family-Work Interaction
		%	%	%
Burnout ^a^	Yes	76.3%	0.0%	23.7%
	No	30.9%	10.2%	58.9%
Pearson’s Chi-squared		77.889		
Asymptotic sig. (bilateral)		0.000 *		
Emotional exhaustion ^b^	High	54.4%	7.0%	38.6%
	Medium	19.8%	14.0%	66.2%
	Low	21.6%	7.6%	70.8%
Pearson’s Chi-squared		130.090		
Asymptotic sig. (bilateral)		0.000 *		
Depersonalisation ^b^	High	47.6%	7.1%	45.3%
	Medium	36.8%	10.0%	53.3%
	Low	25.2%	10.4%	64.4%
Pearson’s Chi-squared		40.254		
Asymptotic sig. (bilateral)		0.000 *		
Personal realisation ^b^	High	54.2%	7.9%	37.9%
	Medium	38.3%	13.3%	48.5%
	Low	21.4%	7.1%	71.5%
Pearson’s Chi-squared		94.629		
Asymptotic sig. (bilateral)		0.000 *		

^a^ Grouping variable: burnout; ^b^ grouping variables: emotional exhaustion, depersonalization and personal realization; * *p*-value of the Chi-squared test.

**Table 4 ijerph-19-03717-t004:** Group statistics and Pearson’s Chi-squared test for marital status, parenthood and contact with SARS-CoV-2.

		Marital Status					Parenthood		Contact with SARS-CoV-2	
		Married	Single	Divorced	Widow	With a Partner	Yes	No	Yes	No
		%	%	%	%	%	%	%	%	%
Dependent variables ^a^										
EE	High	48.7%	33.9%	39.2%	0.0%	53.7%	45.4%	38.6%	49.6%	34.3%
	Med.	28.9%	36.7%	45.1%	28.6%	19.0%	29.2%	33.4%	21.7%	42.5%
	Low	22.4%	29.4%	15.7%	71.4%	27.2%	25.4%	27.9%	28.7%	23.2%
Pearson’s Chi-squared		67.688					4.360		51.059	
Asymptotic sig. (bilateral)		0.000 *					0.113 *		0.000 *	
DP	High	33.7%	26.9%	23.5%	28.6%	33.3%	32.3%	28.2%	38.3%	21.1%
	Med.	21.5%	24.7%	35.3%	0.0%	44.9%	24.0%	29.6%	24.0%	28.7%
	Low	44.8%	48.4%	41.2%	71.4%	21.8%	43.6%	42.2%	37.7%	50.1%
Pearson’s Chi-squared		60.043					4.154		34.273	
Asymptotic sig. (bilateral)		0.000 *					0.125 *		0.000 *	
PR	High	23.0%	23.4%	39.2%	0.0%	36.1%	25.9%	24.1%	28.9%	20.5%
	Med.	24.3%	38.3%	41.2%	28.6%	42.2%	30.6%	35.3%	35.6%	28.0%
	Low	52.6%	38.3%	19.6%	71.4%	21.8%	43.5%	40.5%	35.4%	51.5%
Pearson’s Chi-squared		76.535					2.373		26.298	
Asypmtotic sig. (bilateral)		0.000 *					0.305 *		0.000 *	

^a^ Grouping variables: emotional exhaustion (EE), despersonalisation (DP) and personal realisation; * *p*-valueof the Chi-squared test.

**Table 5 ijerph-19-03717-t005:** Correlation of Spearman’s Rho for emotional exhaustion and depersonalization with service time.

Spearman’s Rho		Service Time	Emotional Exhaustion	Depersonalisation
Service time	Correlation coefficient	1.000	0.149 **	0.014
	Sig. (bilateral)	.	0.000	0.662
	N	128	128	128
Emotional exhaustion	Correlation coefficient	0.149 **	1.000	0.363 **
	Sig. (bilateral)	0.000	.	0.000
	N	128	128	128
Depersonalisation	Correlation coefficient	0.014	0.363 **	1.000
	Sig. (bilateral)	0.662	0.000	.
	N	128	128	128

** Correlation is significant at 0.01 (bilateral).

## Data Availability

The data presented in this study are available on request from the corresponding author. The data are not publicly available due to privacy restrictions.

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
