# Peer review of "Work–Family Interaction, Self-Perceived Mental Health and Burnout in Specialized Physicians of Huelva (Spain): A Study Conducted during the SARS-CoV-2 Pandemic"

_ijerph, 2022, doi:10.3390/ijerph19063717_

Round 1

Reviewer 1 Report

This manuscript by Gago-Valiente al et al: “Work-Family interaction, self-perceived mental health and burnout in specialized physicians of Huelva (Spain): a study conducted during the SARS-CoV-2 pandemic” is based on self-analysis research that study the mental health of medical staff of Huelva (Spain), during SARS-CoV-2 pandemic. The studies that investigate the mental health in the circumstances of COVID19 pandemic are currently very popular, but new explorations are always interesting. However, this manuscript seems unaccomplished, and some important items have been left vague. The figures and tables are insufficient, and the data are not presented clearly. The descriptions of the figures are poor, only the title (frequently half-done is included). That have to be improved.

Minor points:

  • Lines 88-89: “During the last two years several reports have been published about the effects of the SARS-CoV-2 pandemic on the mental health of healthcare workers [17].” – I cannot find study of COVID19 in this reference. Authors mentioned “several reports”, that are not cited here. Furthermore, the following sentence: “However, all the information is important, as it allows a more comprehensive analysis of this problem: the impact of the pandemic on the well-being of professionals who are on the front line of ” is not clear.
  • Material and methods (2.1. Participants): The marital status is presented as a graph in a Figure 1. There is an confusion: “Single”, “Divorced” and “Widowed” are all “Single”, that all together means “Without a partner” with whom they live? And “Married” and “With a partner” are all “With a partner”- in the same household? In this section all groups of participants that are included (especially men and women, but also the parental status) should be presented as diagram (or in a table). The numbering of percentages only in the text is hard to be followed. This is a general problem in this manuscript.
  • Material and methods: Figure 2. The legend is almost invisible, as well the font of the numbers and letters in the graph.
  • The authors cite some questionnaires, and explain the others. That is not sufficient. All questionnaires (as experimental methods) have to be included in the manuscript, at least as a supplemental material.
  • Material and methods: lanes 183-197: The evaluation of the level of affectation: It is explained in a little bit confusing way.

“…based on the three subscales or dimensions, the results are categorized in high, medium, and low levels, and the scores are indicated below:

  • Emotional exhaustion: high (≥27), medium (19 – 26) and low (≤18).
  • Depersonalization: high (≥10), medium (6 – 9) and low (≤5).
  • Personal realization: high (≥40), medium (34 – 39) and low (≤33).“

The authors claim: “The lower the score, the greater the level of affectation for emotional exhaustion and depersonalization”. Does it mean that “the levels” (high/medium/low) indicate the “levels of affectation”, which score should be low for high affectation and vice versa? If I have understood well, it means, in emotional exhaustion, that the score of ≤18 means high level of affectation. The low is only the score? Therefore, the level for emotional exhaustion is expected to be low (≥27), medium (19 – 26) and high (≤18)? If that's true, the same is valid for Depersonalization and Personal realization.

  • Discussion – lines: 403-404: „The results present a higher incidence in the positive interaction of family over work in both men and women.” – I cannot find those data clearly presented. The authors should refer to the results that are stated.

Major points:

  • The descriptions of the results are insufficiently explained and poorly presented. I understand that detailed analysis of the results takes place in discussion. However, short sentence of introduction, and at least, a short comment is required.
    • Work family interactions – the section is explained in only one sentence. The ratios are mentioned for the whole sample: negative work-family interaction, positive work-family interaction, and positive family-work interaction. What is with negative family-work interaction? It is also possible. All results have to be presents as tables or graphs, and furthermore explained in the text.
    • Probable non-psychotic psychiatric pathologies (GHQ-12) – The description of Figure 3 is insufficient. Furthermore, the authors claim that “most of the participants did not present possible non-psychotic psychiatric pathologies.” Is true, but, still, more that 45% of both men and woman have presented possible non-psychotic psychiatric case? That should not be neglected. Finally, authors claim that “The percentage of affectation of women was slightly higher than that of men”. The difference is only 53% for woman and 54,5 for men, that is negligible.
    • Prevalence of burnout, high emotional exhaustion, high depersonalization and low personal realization – the results are again not commented. The Table 1 have to be arranged. The term “Percentages (%)” is not explained. Furthermore, the burnout (that is important parameter for this study) has been presented in only 5,1% of participants. The authors should discuss the relevance of results on that small sample of participants with burnout considering following investigation of different groups and correlations in participants with burnout (e.g. Sections 3.4, 5).
    • Correlation of work-family interaction with sex, situations of contact with SARS-CoV-2, burnout, emotional exhaustion, depersonalization, and personal realization – I cannot find the analysis of man and woman separately for negative/positive work-family/family-work. It is important for the goals of this study. Furthermore, negative family-work interaction is not presented in table 2. Why?
    • Correlation of burnout and its dimensions with sex, marital status, contact with situations of SARS-CoV-2 and parenthood – The section is extensive, but consisted mostly of data enumeration.
  • Lines 294-313 – graphs or tables have to be included for: (1) the analysis of the relationship between burnout and sex (lines 295-305), (2) the relationship between burnout and marital status (lines 306-309), and (3) Dependence between contact with situations of SARS-CoV-2 and burnout (lines 310-313).
  • The sex has not been analysed in table 3. Why?
    • Correlation of the results in GHQ-12 (possible or non-possible non-psychotic psychiatric case) with burnout, emotional exhaustion, depersonalization, personal realization and contact with situations of SARS-CoV-2 – Figure or table that represent result have to be included
    • Correlation of the result in GHQ-12 with sex, marital status and parenthood – Explanation of the results is missing. Interestingly, the majority of participants “with a partner” belong to the category “non-probable non-psychotic case”, and the result was similar in the category of “divorced” participants. Otherwise, the ratio of “non-probable non-psychotic case” and “probable non-psychotic case” was similar in the group of married participants.

Reviewer 2 Report

The  study aimed to identify the interactions between the work and family environments, as well as to analyze self- perceived mental health and burnout in physicians during the COVID-19 pandemic in Huelva (Spain). The study did consider different sociodemographic variables, concluding that the medical staff of Huelva who had been in contact with situations of SARS-CoV-2 in their work  presented worse indicators of mental health and greater negative interaction of work over  family than those who had not been in contact with these situation.

I feel the topic very interesting and the quality of the presentation and references  are consistent. I would improve the introduction with some international data and studies.

I suggest a language review , possibly by a native speaker.

LINE 57: please reformulate the sentence"The resulting professional discomfort..."

LINE 70-72 : please add a list form

LINE 73: replace specific and special

LINE 99: please be clarify the sentence

LINE 164:  please edit the English form

LINE 426: English editing "The conclusions of other works [35 – 37] agree with many of the findings of the pre-426 sent study"

please check the reference number 44,

Round 2

Reviewer 1 Report

The authors have impoved the manuscript according to the recommendations. I suggest the acception of thist study in present form.